# Toward a methodology for evaluating DNA variants in nuclear families

**Dustin B. Miller**[1], **Reid Robison**[1,2], **Stephen R. Piccolo**[1]*

**1** Department of Biology, Brigham Young University, Provo, UT, United States of America, **2** Department of Psychiatry, University of Utah, Salt Lake City, UT, United States of America

* stephen_piccolo@byu.edu

**Data Availability Statement:** Aligned BAM files from this study are available from the European Genome-Phenome Archive (accession: EGAS00001005321).

**Funding:** The author(s) received no specific funding for this work.

## Abstract

The genetic underpinnings of most pediatric-cancer cases are unknown. Population-based studies use large sample sizes but have accounted for only a small proportion of the estimated heritability of pediatric cancers. Pedigree-based studies are infeasible for most human populations. One alternative is to collect genetic data from a single nuclear family and use inheritance patterns within the family to filter candidate variants. This approach can be applied to common and rare variants, including those that are private to a given family or to an affected individual. We evaluated this approach using genetic data from three nuclear families with 5, 4, and 7 children, respectively. Only one child in each nuclear family had been diagnosed with cancer, and neither parent had been affected. Diagnoses for the affected children were benign low-grade astrocytoma, Wilms tumor (stage 2), and Burkitt's lymphoma, respectively. We used whole-genome sequencing to profile normal cells from each family member and a linked-read technology for genomic phasing. For initial variant filtering, we used global minor allele frequencies, deleteriousness scores, and functional-impact annotations. Next, we used genetic variation in the unaffected siblings as a guide to filter the remaining variants. As a way to evaluate our ability to detect variant(s) that may be relevant to disease status, the corresponding author blinded the primary author to affected status; the primary author then assigned a risk score to each child. Based on this evidence, the primary author predicted which child had been affected in each family. The primary author's prediction was correct for the child who had been diagnosed with a Wilms tumor; the child with Burkitt's lymphoma had the second-highest risk score among the seven children in that family. This study demonstrates a methodology for filtering and evaluating candidate genomic variants and genes within nuclear families that may merit further exploration.

## Introduction

Genome-wide association studies (GWAS) have identified novel cancer susceptibility loci for some pediatric cancer types [1–4]. However, in GWAS studies, the research participants are typically unrelated, and rare variants are not easily identified due to a lack of statistical power

**Competing interests:** The authors have declared that no competing interests exist.

[5,6]. Accordingly, these studies may fail to discover rare variants that are responsible for some of the missing heritability of complex diseases [6]. Family-based designs are sometimes able to identify rare candidate variants when related, affected individuals have the same variant in common [7]. For example, in co-segregation studies, researchers analyze related individuals for variants that segregate with affected status. Perhaps the ideal co-segregation design involves sequencing the DNA of individuals spanning many generations; however, these samples are difficult to obtain for most human populations. A simpler, yet more feasible, family-based design involves sequencing trios (an affected child and both parents). This approach enables investigators to identify the parent of origin for most variants under investigation and facilitates identification of compound-heterozygous variants (when a proband inherits a variant from each parent in the same gene at different loci) and *de novo* variants (when both parents lack a variant that is found in a proband) [8]. Both of these variant types contribute to pediatric cancers and are relatively understudied [9,10].

An extension of the trio-based design is to study nuclear families in which DNA has been collected from both parents and two or more children. In 2010, Roach *et al.*, used this approach in combination with next-generation sequencing to profile a nuclear family of four [11,12]. Both children had Miller syndrome and primary ciliary dyskinesia, which are Mendelian disorders [11]. Analyzing whole genome sequencing (WGS) data across all members of the nuclear family, they narrowed the candidate genes for these diseases to four. More recently, a 2016 study by Stittrich *et al.* analyzed WGS data of five nuclear families, plus some extended family members, to identify variants that conferred a risk for inflammatory bowel disease (IBD) [13]. Nuclear family sizes (excluding extended family members) ranged from four to eight individuals, with at least one child having been diagnosed with IBD. Extended family members had been diagnosed with IBD or were suspected to have IBD. Using identity-by-descent of affected and unaffected individuals from nuclear and extended family members in conjunction with variant- and gene-level filters, they identified rare, novel variants that conferred a risk for disease development. Protein modeling and a luciferase reporter assay were used to test potential effects of the top candidate variant. These studies highlight that sequencing many members of a nuclear family can provide insights that would not be provided through a trio-based approach. In addition to identifying shared variants among siblings with the same disease, such designs can provide a clearer understanding of the genetic underpinnings of a disease, even when only one child has been diagnosed with the disease. For example, sequencing the DNA of all members of a nuclear family may allow investigators to eliminate certain variants from being considered as "disease-causing" when one or more healthy children have the same variant as the proband. On the other hand, if a variant is considered disease-causing in the proband and this same variant is found in a sibling, it may indicate that the non-proband child is susceptible to developing the disease in the future; this methodology could be used for prenatal screening.

The state of Utah has the highest average birth rate (14.9 per 1,000 women) in the United States, much higher than the national average (11.6) [14]. Anecdotally, we had observed that families with 4 or more children are common in this population. We performed a pilot study to evaluate the potential to use DNA variation from unaffected children in a relatively large nuclear family as a guide when filtering candidate variants for an affected child in the same family. Via collaboration with a nonprofit organization that supported families affected by pediatric cancer, we identified three Utah families in which a single child had been diagnosed with some form of pediatric cancer and for which at least four children had been born to the same parents. These families had 5, 4, and 7 children, respectively, for a total of 22 individuals (including parents) (Fig 1). Diagnoses for the affected children were benign low-grade astrocytoma, Wilms tumor, and Burkitt's lymphoma, respectively. Low-grade astrocytomas develop

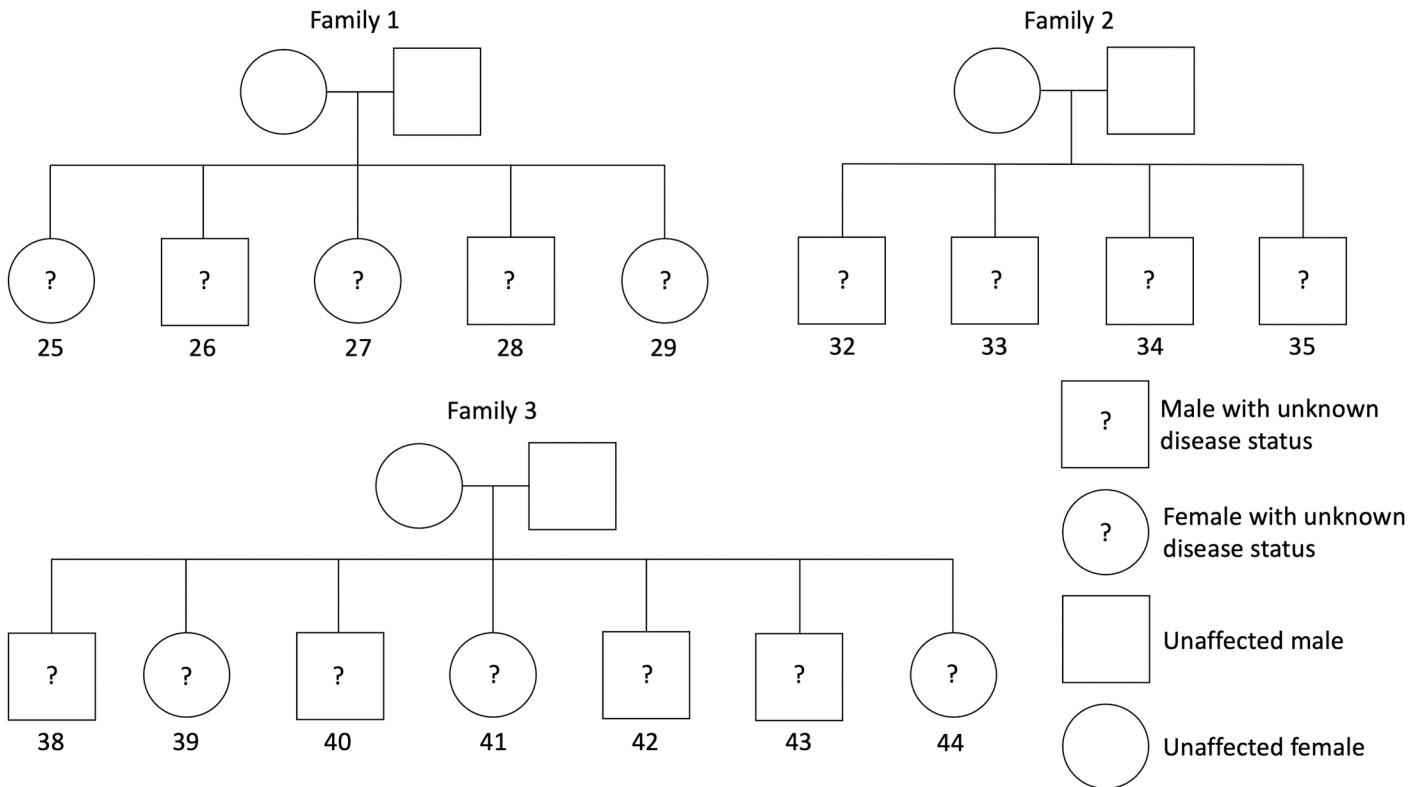

**Fig 1. Structure of each nuclear family.** Each diagram in this figure represents one family from our study. In each diagram, shapes at the top represent parents; shapes at the bottom represent children. Numbers below children are sample identifiers. These identifiers do not necessarily correspond to birth order or any other factor. One child in each family had been diagnosed with pediatric cancer, but the disease status of all children was unknown to the primary author until after disease-history predictions were made. This unknown status is represented using question marks.

in the central nervous system and are the most common brain tumor type in children [15]. Wilms tumor, one of the most common solid tumors occurring in children, develops in the kidney [16]. Burkitt's lymphoma is a type of non-Hodgkin lymphoma that originates from B-cells [17]. For each nuclear family, we generated linked-read WGS data for each individual from saliva samples.

By sequencing all members of these nuclear families, we hoped to identify DNA variants that were candidates as causal variants for each affected child. Two ways of validating such candidates include observation in subsequent familial generations and lab-based functional testing; however, waiting to observe subsequent generations is infeasible for families with young children, and designing functional tests can be expensive and time consuming when evaluating multiple variants of unknown clinical significance on an individual-sample basis [18]. As an alternative form of validation, we evaluated whether we could use candidate DNA variants within a nuclear family to make correct predictions in retrospect about which child had been diagnosed with cancer. By random chance, it would be difficult to correctly predict the phenotypic status of all children in our study. The probability of predicting the correct affected child by random chance for the individual families would be 1 in 5 (0.20), 1 in 4 (0.25), 1 in 7 (0.143), respectively. Using the general multiplication rule, the probability of making correct predictions for all three families was 0.00715. Accordingly, if we were able to make such predictions correctly, it would provide evidence that using DNA variation from unaffected members of the same nuclear family merits further exploration as a guide for

variant interpretation. To test this idea, we used a blind-study format. The primary author (DBM) analyzed the genomic data without any knowledge of which child in each nuclear family had been diagnosed with pediatric cancer. Based on variant minor allele frequencies, deleteriousness scores, functional-impact annotations, gene-disease association scores, and disease-likelihood scores, DBM made predictions about which child had been diagnosed with cancer in each family. He then deposited a summary of our research protocol and the individual-level predictions in a preregistration repository on the Open Science Framework website [19] (doi: 10.17605/OSF.IO/89Y67). After preregistration, SRP revealed the prior diagnosis status of each child, and together they assessed the accuracy of the predictions.

## Results

We generated germline, WGS data for three nuclear families (Fig 1). Family 1 had 5 children, whom we labeled with sample identifiers 25–29; one of these children had been diagnosed earlier in life with a benign low-grade astrocytoma. Family 2 had 4 children, whom we labeled with sample identifiers 32–35; one of these children had been diagnosed previously with a Wilms tumor. Family 3 had 7 children, whom we labeled with sample identifiers 38–44; one of these children had been diagnosed previously with Burkitt's lymphoma. Author DBM used the genetic data and associated annotations to make a prediction for each family about which child had been diagnosed with cancer.

### Summary of identified variants

Variant data included phased single-nucleotide variants (SNPs), mid-scale deletions (indels), and large-scale structural variants (SVs). The total number of passing variants (quality score >= 20 and filter classification of "PASS") was similar for most samples (Fig 2). For each patient, we identified simple-heterozygous, homozygous-alternate, and compound-heterozygous variants and filtered the variants based on global minor allele frequencies (MAF), Combined Annotation Dependent Depletion (CADD) scores [20], impact severity, and whether the variants were exonic (see Methods; Fig 3). For each of these variant types, we also examined whether each variant was de novo (undetected in either parent) (Fig 4). For Family 1, across all children, we identified 33 potentially damaging, simple-heterozygous variants and 1 potentially damaging homozygous-alternate variant. Eight of the simple-heterozygous variants were de novo. Of the potentially damaging variants identified in Family 2, 20 were simple heterozygous, 1 was compound heterozygous, and 1 was de novo homozygous alternate. Of the simple-heterozygous variants, 6 were de novo. One of the heterozygous variants contributing to the compound-heterozygous variant was de novo. Across all children in Family 3, we identified 30 potentially damaging, simple-heterozygous variants and 7 potentially damaging, homozygous-alternate variants; 11 of the simple-heterozygous variants and 3 of the homozygous-alternate variants were de novo. The number of SNPs, indels, and SVs varied within families. For example, the number of SNPs per child in Family 3 ranged from 5 to 9. Every child in Family 3 had at least one indel, but only 2 of the 7 children had at least one SV. We observed similar trends for Families 1 and 2.

### Variants in common among siblings

We assessed the potentially damaging variants and the gene(s) in which the variants were identified for commonality among siblings (Table 1). We assumed that if multiple children within a family had a potentially damaging variant in the same gene, then the gene was less likely to influence disease development. However, we allowed for the possibility that two children in a family might have a damaging variant. We supposed that if such a variant were shared by the

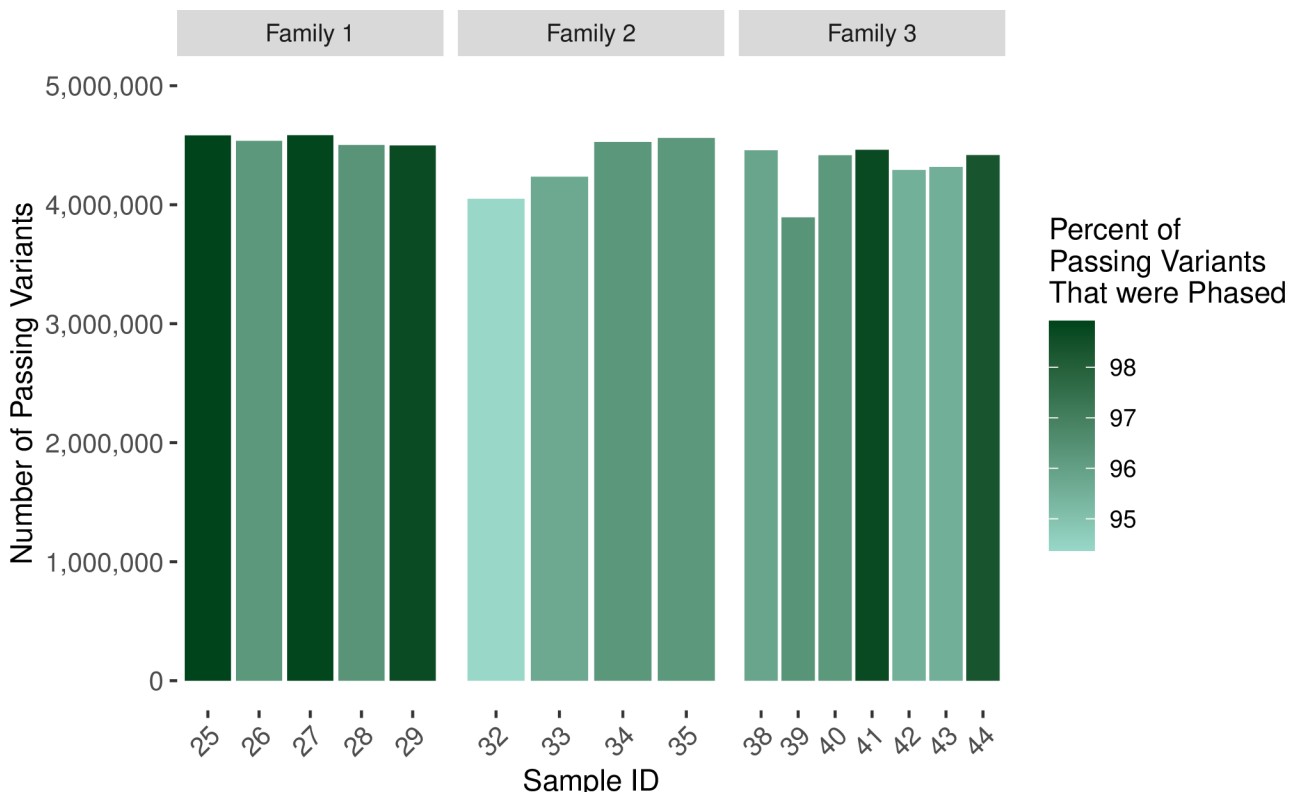

**Fig 2. The number of passing variants output from *Long Ranger* and the percentage of those passing variants that were phased.** Passing variants were those that had a "PASS" in the filter column of the VCF and a quality score $> = 20$. These numbers are summarized over all variant categories. In order of the samples listed in the figure, the number of total variants output from *Long Ranger* per sample was $5.89 \times 10^6$, $5.95 \times 10^6$, $6.11 \times 10^6$, $6.00 \times 10^6$, $6.22 \times 10^6$, $5.79 \times 10^6$, $5.80 \times 10^6$, $6.12 \times 10^6$, $6.02 \times 10^6$, $6.07 \times 10^6$, $5.72 \times 10^6$, $5.83 \times 10^6$, $6.58 \times 10^6$, $6.25 \times 10^6$, $6.19 \times 10^6$, $6.40 \times 10^6$.

affected child and an unaffected sibling, the variant might influence tumorigenesis later in life in the unaffected sibling. Another possibility is that epistasis may have occurred between this damaging variant and another damaging variant in the affected child but the interacting variant was not present in the unaffected sibling. This filtering step reduced the number of candidate variants to 17, 17, and 23 in 15, 16, and 22 mutated genes for Family 1, Family 2, and Family 3, respectively (Tables 1 and S1). Most of the mutated genes were private within each family. For example, in Family 1, 10 mutated genes were unique to any single child within the family, while the remaining 5 mutated genes were shared by two children.

### Gene rankings and pediatric-cancer predictions

We evaluated each mutated gene using *VarElect*, which provides details from the literature about gene-phenotype associations. We used this information to rank genes and score each patient (see Methods). A complete list of genes with potentially damaging variants and the number of samples with a potentially damaging variant in each gene is provided in S1 Table.

The top-five ranked genes for Family 1 (benign low-grade astrocytoma) were *FAM8A1*, *ACADS*, *KRT76*, *HLA-DRB1*, and *TTBK2* (S2 Table). Based on our scoring methodology, we predicted Sample 26 to have been the child diagnosed with cancer (Table 2). Sample 26 had two simple-heterozygous SNPs in *FAM8A1* (both on the same chromosome) and one simple-heterozygous SNP in *ACADS*. These SNPs resulted in early stop sequences in these genes. While neither of the genes have directly been associated directly with low-grade astrocytomas,

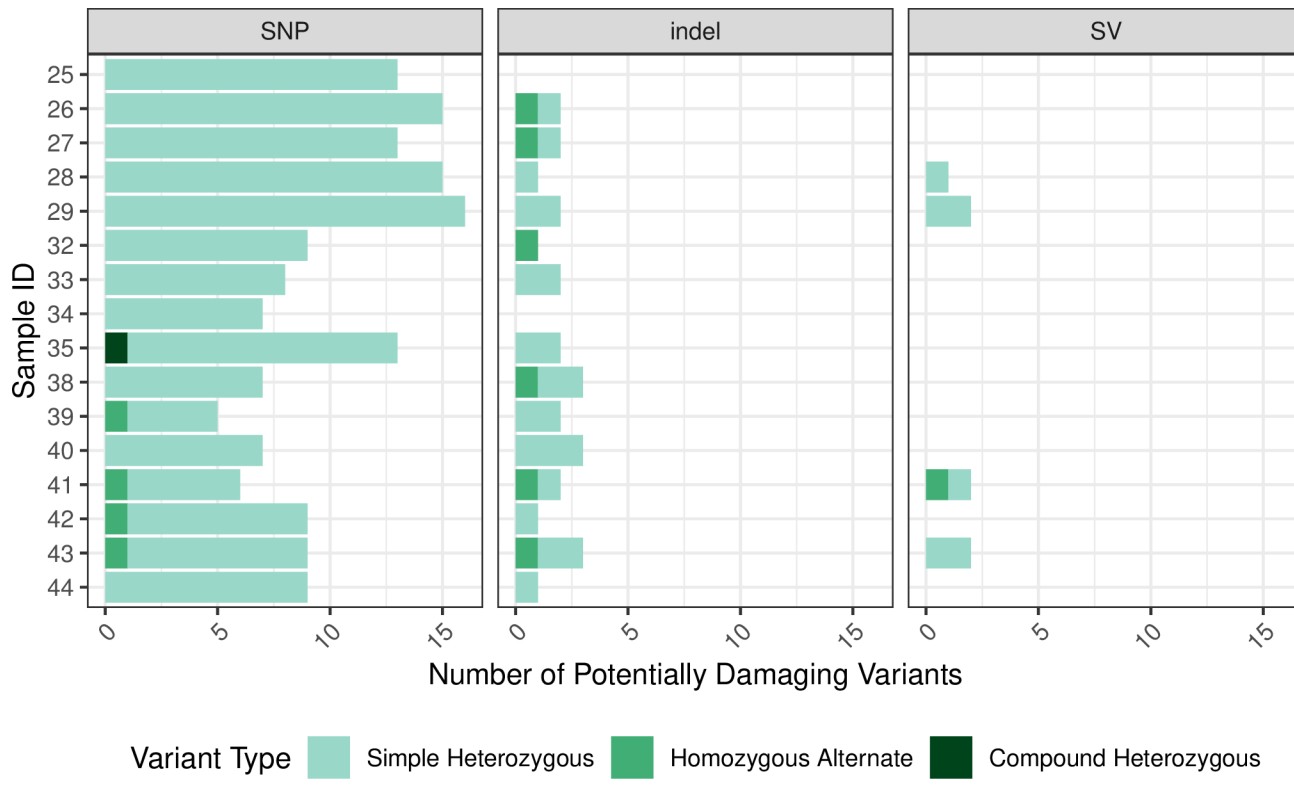

**Fig 3. Summary of potentially damaging variant types.** This figure shows the number of potentially damaging variants for each variant type (simple heterozygous, homozygous alternate, and compound heterozygous) and category (SNP, indel, and SV) in each sample. Potentially damaging variants met specific MAF, CADD and impact severity thresholds (see Methods).

they are involved in important cellular processes. FAM8A1 is a membrane protein that helps assemble the HRD1 complex, which is part of the ubiquitin-proteasome-dependent process of endoplasmic-reticulum-associated degradation (ERAD) [21]. This process targets misfolded proteins, which are then degraded. The ACADS protein is a flavoenzyme that is involved in fatty acid catabolism in the mitochondria [22]. A deficiency in ACADS production can inhibit some fats from being converted to energy [23].

For Family 2 (Wilms tumor), the top-five ranked genes were *FAM8A1*, *TRPM3*, *PAH*, *PCK1*, and *PCK2* (S3 Table). We predicted Sample 35 to have been the child diagnosed with cancer (Table 3). This child had a de novo, simple-heterozygous deletion in *FAM8A1*, leading to a frameshift; it also had simple-heterozygous SNPs in *TRPM3* and *PAH*, leading to early stop sequences in each gene. Interestingly, the top-ranked gene (FAM8A1) was identical to the top-ranked gene for Family 1; however, as with low-grade astrocytomas, this gene has not been associated directly with Wilms tumors. The TRPM3 protein functions as an ion channel to allow calcium to pass through the surface of a cell [24]. Some mutations in this gene can create an overactive ion channel. Altered expression of *TRPM3* has been observed in glioblastoma [25] but not in Wilms tumors. PAH is an enzyme involved in phenylalanine catabolism [26]. Inherited, autosomal recessive defects in *PAH* can lead to phenylketonuria, a disease that causes physical and mental abnormalities.

For Family 3 (Burkitt's lymphoma), *TNNT3*, *SIRPB1*, *TRMT1*, *ITGB4*, and *DCHS1* were the top-five ranked genes (S4 Table). Sample 43 had the highest score and was predicted to be the child diagnosed with cancer (Table 4). This child had a de novo, simple-heterozygous

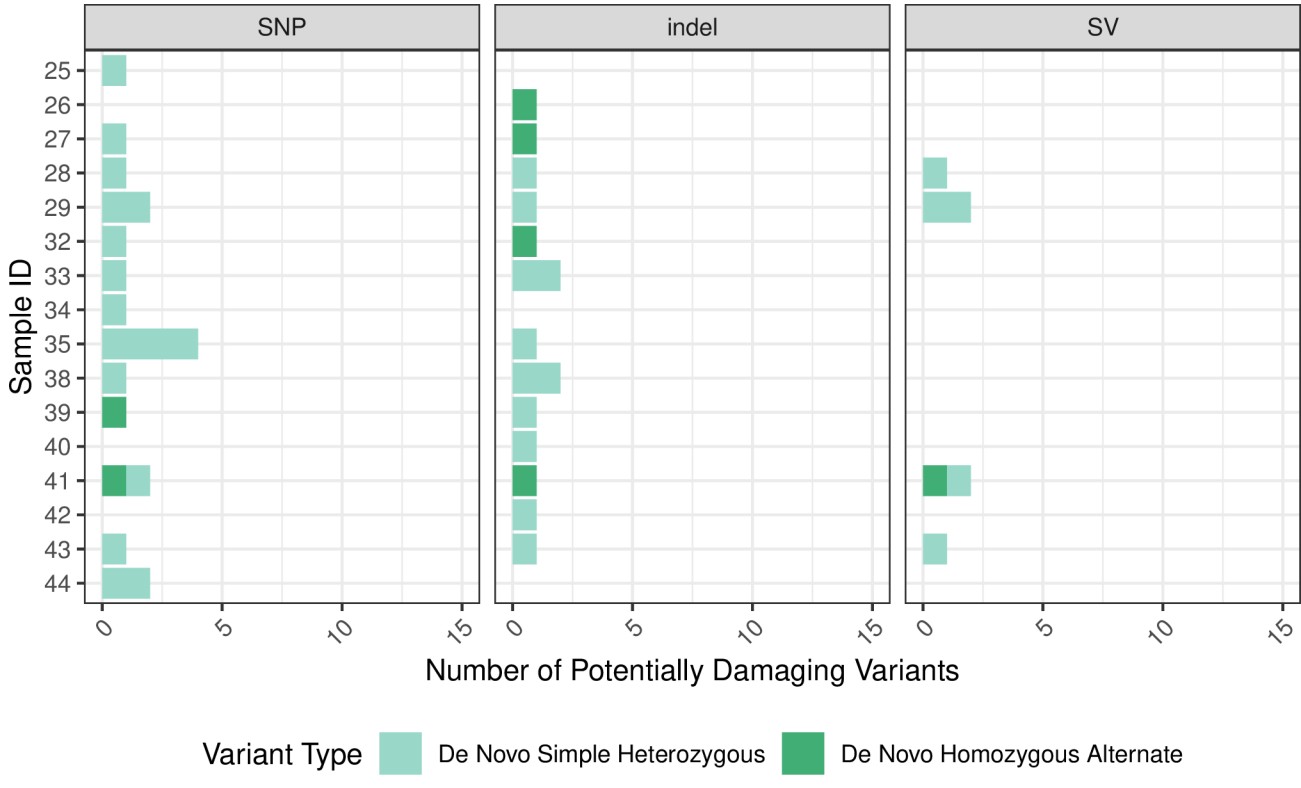

**Fig 4. Summary of de novo variants.** This figure shows the number of variants that were de novo for each variant category for each sample.

structural variant in *TNNT3*, leading to a transcript ablation, and a de novo, simple-heterozygous SNP in *ITGB4*, creating an early stop sequence. Neither of these genes have been directly related to Burkitt's lymphoma. *TNNT3* is a troponin T isoform that helps to control muscle contraction through calcium regulation [27]. The *ITGB4* gene codes for an integrin transmembrane receptor involved in extracellular matrix interactions [28]. Altered expression of *ITGB4* has been observed in various cancers including lung and breast [29,30].

## Evaluation of pediatric-cancer predictions

After author DBM publicly recorded predictions for each family, author SRP revealed the cancer status of each child. We found that one of the three predictions was correct—Sample 35 of Family 2 had been diagnosed with a Wilms tumor. This sample had unique variants (not shared with other siblings) in the top two ranked genes (*FAM8A1*, and *TRPM3*) and shared a simple-heterozygous SNP with Sample 34 in the third ranked gene (*PAH*). Family 2 was the smallest of the three families; thus the probability of making a correct prediction by random chance was highest for this family. In Family 1, Sample 25 had been diagnosed with a benign low-grade astrocytoma but was ranked fourth out of five children using our methodology. This individual had a simple-heterozygous SNP in *HLA-DRB1*, the 3rd-ranked gene (tied with *KRT76*); no other child in the family shared this variant. The HLA-DRB1 protein plays a role in the immune system by presenting extracellular antigens to helper T cells [31]. Variants in *HLA-DRB1* have been associated with multiple sclerosis [31]. In Family 3, Sample 41 was revealed to have had Burkitt's lymphoma and was ranked second out of seven children using our methodology. This individual had a de novo, homozygous structural variant in *SIRPB1*, the 2nd overall ranked gene for Burkitt's lymphoma; no other child in the family shared this

**Table 1. An overview of variants in common among children within each nuclear family.**

|  | Number of children | Number of shared variants | Number of shared genes |
|---|---|---|---|
| **Family 1** | 1 | 12 | 10 |
|  | 2 | 5 | 5 |
|  | 3 | 7 | 6 |
|  | 4 | 6 | 6 |
|  | 5 | 3 | 3 |
| **Family 2** | 1 | 12 | 11 |
|  | 2 | 5 | 5 |
|  | 3 | 4 | 4 |
|  | 4 | 2 | 2 |
| **Family 3** | 1 | 15 | 14 |
|  | 2 | 8 | 8 |
|  | 3 | 5 | 4 |
|  | 4 | 5 | 4 |
|  | 5 | 1 | 2 |
|  | 6 | 0 | 0 |
|  | 7 | 0 | 0 |

We excluded variants that were present in more than two children in a given nuclear family. This table shows the number of variants in common among a specified number of siblings. For example, in Family 1, 12 variants (in 10 genes) were unique to a single child; 5 variants (in 5 genes) were shared between two children. These numbers reflect the number of "potentially damaging" variants—including simple-heterozygous, compound-heterozygous, and homozygous-alternate variants—that met specific MAF, CADD and impact-severity thresholds (see Methods).

variant. *SIRPB1* is involved in cell signaling as a transmembrane glycoprotein receptor and may be associated with immunodeficiencies [32,33].

## Discussion

This study contributes a novel methodology for estimating the translational importance of DNA variants in a given family. Unlike other designs that use trios or quads or that focus primarily on affected family members, we collect data mostly from unaffected individuals, including parents and siblings. Despite the inherent small sample size of comparing individuals within a nuclear family, family members are often exposed to environmental conditions, cultural practices, and dietary habits that are more similar to each other than to the general population, thus potentially reducing confounding factors that could bias an analysis. Although we were not able to demonstrate that our predictions were more accurate than expected by random chance, we have introduced a methodology for identifying candidate variants that can be expanded or refined. Furthermore, we have shared a unique dataset with the community; we are unaware of other WGS datasets from nuclear families with as many as seven children.

Different from population-based, genetic studies that seek to identify variants or genes of interest, we attempted to predict proband status for an individual child in a given nuclear family. Our approach may provide insight into variants and genes that have influenced disease development within a given family but not necessarily for the broader population. Different from studies that use genotypic markers to derive risk scores based on population-based frequencies [34], we used inheritance patterns within nuclear families, supplemented by variant annotations and literature-based evidence. Accordingly, our approach is a mixture of quantitative and qualitative methods. Quantitatively, we used global minor allele frequencies,

**Table 2. Top five candidate genes for Family 1 (benign low-grade astrocytoma).**

| Gene | Gene Rank | Gene Score | Sample (type of variant(s); number of potentially damaging variants) | Sample (gene score points awarded) |
|---|---|---|---|---|
| FAM8A1 | 1st | 5 | 26 (het SNP; n = 2), 28 (het SNP) | *26 (2.5), 28 (2.5) |
| ACADS | 2nd | 4 | 26 (het SNP) | *26 (4) |
| KRT76 | 3rd (tie) | 3 | 29 (het SNP) | 29 (3) |
| HLA-DRB1 | 3rd (tie) | 3 | 25 (het SNP) | ^25 (3) |
| TTBK2 | 5th | 1 | 27 (het del), 29 (het del) | 27 (0.5), 29 (0.5) |

We ranked genes based on the average gene-disease connection and disease likelihood scores (S2 Table). The gene scores were equally divided among the samples when more than one sample shared a variant in that gene.

"*" indicates which child was predicted to have had cancer;

"^" indicates the child who had cancer. Sample 26 had simple-heterozygous, SNP variants in the top 2 genes and was predicted to have had benign low-grade astrocytoma based on having the highest aggregate gene score of 6.5. Abbreviations: het = simple heterozygous. del = deletion.

deleteriousness scores, functional-impact annotations, prior disease associations, and co-occurrence among siblings to inform variant- and gene-level filtering. However, to derive a single prediction per family, qualitative judgment was necessary to define filtering thresholds and to combine multiple lines of evidence. Some combinations of thresholds and evidence aggregation may have resulted in accurate predictions for all three families, while many other combinations would not. Therefore, our question was not whether it would be possible to identify such patterns but whether we could make accurate predictions without knowing proband status *a priori*. Accordingly, we blinded the first author and made our predictions publicly available before proband stata were revealed to the first author.

Only one of the three predictions that we made was correct, suggesting that germline variation and currently available ancillary information provide insufficient information to make accurate predictions and/or that our prediction methodology requires further refinement. Furthermore, tumorigenesis often results from at least one mutation inherited in germline cells and at least one mutation that has occurred sporadically in somatic tissue [35]. Our methodology does not account for somatic mutations nor epigenetic factors, such as aberrant DNA methylation, that may play a role in tumor development [36].

Our use of linked-read WGS enabled us to estimate phase reliably and thus to identify de novo and compound-heterozygous variants. Sequencing a large number of children in each family allowed us to filter out variants that were common among multiple siblings, potentially excluding genes that are less likely to contribute to pediatric cancer development. By using *VarElect* to evaluate known gene-disease connections and disease likelihood, we were able to

**Table 3. Top five candidate genes for Family 2 (Wilms tumor).**

| Gene | Gene Rank | Gene Score | Sample (type of variant(s); number of potentially damaging variants) | Sample (gene score points awarded) |
|---|---|---|---|---|
| FAM8A1 | 1st | 5 | 35 (de novo het del) | **35 (5) |
| TRPM3 | 2nd | 4 | 35 (het SNP) | **35 (4) |
| PAH | 3rd (tie) | 3 | 34 (het SNP), 35 (het SNP) | 34 (1.5), **35 (1.5) |
| PCK1 | 3rd (tie) | 3 | 33 (de novo het del) | 33 (3) |
| PCK2 | 5th | 1 | 33 (de novo het del) | 33 (1) |

We ranked genes based on the average gene-disease connection and disease likelihood scores (S3 Table). The gene scores were equally divided among the samples when more than one sample shared a variant in that gene.

"**" indicates a correct cancer prediction. Sample 35 had simple-heterozygous variants (either SNP or deletion) in the top 3 genes and was predicted to have had Wilms tumor based on having the highest individual score of 10.5. Abbreviations: het = simple heterozygous. del = deletion.

**Table 4. Top five candidate genes for Family 3 (Burkitt's lymphoma).**

| Gene | Gene Rank | Gene Score | Sample (type of variant(s); number of potentially damaging variants) | Sample (gene score points awarded) |
|------|-----------|------------|----------------------------------------------------------------------|-------------------------------------|
| *TNNT3* | 1st | 5 | 43 (de novo het SV) | *43 (5) |
| *SIRPB1* | 2nd | 4 | 41 (de novo hom SV) | ^41 (4) |
| *TRMT1* | 3rd | 3 | 44 (de novo het SNP) | 44 (3) |
| *ITGB4* | 4th | 2 | 43 (de novo het SNP) | *43 (2) |
| *DCHS1* | 5th | 1 | 42 (de novo het del) | 42 (1) |

We ranked genes based on the average gene-disease connection and disease likelihood scores (S4 Table). The gene scores were equally divided among the samples when more than one sample shared a variant in that gene.

"*" indicates the child predicted to have cancer and

"^" indicates the actual child with cancer. Sample 43 had simple-heterozygous variants (either SNP or structural variant) in 2 of 5 genes and was predicted to have had Burkitt's lymphoma based on having the highest individual score of 7. Abbreviations: het = simple heterozygous. hom = homozygous. del = deletion. SV = structural variant.

consider complementary types of literature-based evidence. Averaging these ranks helped make our prediction process more objective and prevented us from being influenced unduly by either type of evidence.

After the proband status was revealed for each family, we considered whether differences in sequencing coverage may have biased our analysis. Even though the average sequencing coverage per base varied somewhat across samples, on average, 72.6% of sequenced variants passed our quality filters. The probands were among the samples with the highest number of passing variants and the highest percentage of phased variants. Therefore, if any samples were favored by such a bias, the probands would have been.

We considered alternative strategies that we could have used to make the predictions. One approach would have been to assume that a relatively large number of potentially damaging variants spanning diverse variant classes (SNPs, indels, and SVs) would result in a higher likelihood that a given child would harbor a truly damaging variant. A chi-squared goodness-of-fit test revealed that the total number of potentially damaging variants did not differ significantly among the children in any of the families (p = 0.80, 0.37, 0.87). Anecdotally, in Family 1, the proband had the smallest number of variants (SNPs only); in Family 2, the proband had the largest number of variants (SNPs and indels, including one compound-heterozygous SNP); in Family 3, the proband had the median number of variants (SNPs, indels, SVs). Therefore, basing our predictions on the number and/or diversity of potentially damaging variants would not have led to higher accuracy. Another alternative approach would be to focus on cancer-associated genes. Hundreds of genes have been associated with cancer, mostly in adults [37]. Across all samples, we identified potentially pathogenic variants *RGPD3* and *CRNKL1*. However, for both cancer-associated genes, mutations were not exclusive to the proband in which they occurred.

Having sequencing data from parents enabled us to identify de novo variants, which may have been overlooked otherwise. We identified at least one potentially damaging, de novo variant in every child, across all families (Fig 4). De novo variants develop during gametogenesis and then are transmitted from parent to child; alternatively they can occur during early embryogenesis [8,38]. Therefore, by definition, most de novo variants are rare and should rarely co-occur among siblings within a particular family. Across all families, 23 of the 28 de novo variants that we observed were unique to a single child. It is possible that the remaining, shared variants were a result of sequencing errors, and filtering by co-occurrence among siblings helped mitigate this potential issue, as 4 of the 5 shared variants were excluded after this

filtering step. Overall, genes with de novo variants contributed heavily to our predictions. In particular, for Family 3, all top-ranked genes had a de novo variant that was unique to a single child. For Family 2, three of the five top-ranked genes had a de novo variant.

A key assumption behind our methodology is that if accurate predictions of proband status can consistently be made for nuclear families, the germline variants used to make those predictions may have translational relevance for those families. For example, any such variants that overlap between the affected child and a sibling could be used to indicate cancer risk for the unaffected sibling; having this knowledge could allow parents and clinicians to be more informed and more effectively monitor disease risk. This approach could also be useful for prenatal screening. These potential benefits warrant additional research and further refinement of our methodology.

## Methods

### Data collection

We obtained approval for this study from Brigham Young University's Institutional Review Board (study identifier: X15248). Each nuclear family consisted of a child who had previously been diagnosed with a pediatric tumor, siblings who had not been diagnosed with a pediatric tumor, and both of the affected child's parents. Each affected child was in remission at the time of data collection. Pediatric cancer types included benign low-grade astrocytoma, Wilms tumor (stage 2), and Burkitt's lymphoma. The families had five, four, and seven children, respectively. All family members participated. We obtained signed consent forms from parents and children using age-appropriate consent forms that parents also signed. The consent documents informed participants that de-identified data would be uploaded to a public genomics repository.

We collected a saliva sample from each participant using Oragene 500 collection tubes. After collecting these samples, SRP assigned a random, unique identifier to each participant and labeled the saliva sample with this identifier. A random family identifier was also assigned to the individuals so that family relationships could be determined in a de-identified manner. Personally identifiable information was stored separately from the sequencing data so that these types of data could not be directly linked without having access to both sources. DBM undertook the task of making predictions about which child had been diagnosed with a tumor in each family; he was not involved in recruiting participants, collecting information from the participants, or generating the unique identifiers. At the time when he made the predictions, he was unaware of which child had been diagnosed with a tumor. SRP played no role in making the predictions but was aware of the methods DBM used to make the predictions.

### DNA sequencing and variant calling

The Genomic Services Lab at the HudsonAlpha Institute for Biotechnology extracted DNA from the saliva samples and performed quality checks based on DNA concentrations and bacterial contamination. They performed DNA library preparation using the 10X Genomics Chromium platform. Next, they performed paired-end, whole-genome sequencing using an Illumina HiSeq X system. The reads were 150 bp in length.

Raw, linked-read [39] sequences were assessed for quality, trimmed, and aligned to human reference genome GRCh38 (sub-version 2.1.0) using version 2.1.3 of the *Long Ranger* [40] software. The average coverage per base across the genome for all samples was 35.13. The highest average coverage per base was 43.14, and the lowest average coverage per base was 19.92. These differences among the samples may be due to bacterial contamination or other factors. *Long Ranger* was used to call variants (SNPs, indels, and structural variants). In addition to its

own logic, *Long Ranger* uses algorithms from *BWA* (alignment) [41], *Freebayes* (variant calling) [42], and *Genome Analysis Toolkit* (variant calling) [43]. The results of this process were VCF (Variant Call Format) [44] files.

## Variant preprocessing

For each step of VCF filtering, we used a Docker image (available at https://hub.docker.com/r/dmill903/compound-het-vip) that encapsulated the software tools as well as Python (https://python.org) scripts. The source code is available at https://github.com/dmiller903/PedFam. Some of the scripts used during the variant-filtering process were adapted from CompoundHetVIP [45]. A detailed document showing how the VCF files were processed is available at https://github.com/dmiller903/PedFam/blob/master/code_used_to_analyze_data.pdf.

*Long Ranger* produced 3 VCF files for each sample: 1) phased SNPs, 2) phased mid-scale indels, and 3) phased large-scale SVs. The variants in each of these files were filtered based on quality scores ($> = 20$) and a filter classification of "PASS". *bcftools* [46] *(version 1.9)* was used to combine the SNPs from each sample into a single file, to combine the indels from each sample into a single file, and to combine the SVs from each sample into a single file. This resulted in three different files, each containing data for all 22 samples; these three files were processed separately in all remaining steps.

The SNP file was normalized and left aligned using *vt tools* [47] *(version 2015.11.10)*. *vt tools* only supports VCF files containing single-nucleotide variants; therefore the indel and SV files were not able to be normalized and left aligned. Next, the SNP, indel, and SV files were annotated using *snpEff* [48] *(version 4.3t)*. After annotation, *vcf2db* (https://github.com/quinlan-lab/vcf2db) was used to create databases compatible with *GEMINI* [49] *(version 0.30.2)*. *vcf2db* separated the annotations at each variant position into fields that could be queried; these fields included the impact severity, whether the variant was exonic, etc. This final step of VCF processing produced separate GEMINI databases for SNPs, indels, and SVs.

## Variant filtering

The SNP GEMINI database was queried for simple-heterozygous, homozygous-alternate, compound-heterozygous, and *de novo* variants. When identifying simple-heterozygous and *de novo* variants, we retained variants that had a scaled CADD score greater than 20, a MAF less than 0.01 based on gnomAD [50], an impact severity of "HIGH", and that had been classified as "exonic". We classified these variants as "potentially damaging."

For compound-heterozygous variant identification, we retained variants that had a scaled CADD score greater than 20, an impact severity of "HIGH", and that had been classified as "exonic". We considered these variants to be potentially damaging. For MAF filtering, we required one variant in a given gene to have a MAF smaller than 0.01 but allowed the second variant in the same gene to have a MAF of any value. This allowed for scenarios where a rare allele was paired with a relatively common allele, yet the combined population frequency of the two variants was estimated to be low and thus may be more likely to be disease associated. If a child and either healthy parent had the same compound-heterozygous variant, or if one of the alleles that was part of the compound-heterozygous variant was a homozygous alternate in the parent of origin, it would not likely be disease causing [51]. Thus of the identified compound-heterozygous variants, we retained those that were unique to the child (i.e. not present in either parent). In addition, when identifying compound-heterozygous variants, we excluded from consideration any variants that were homozygous alternate in either parent.

When identifying homozygous-alternate variants, we retained variants that had a scaled CADD score greater than 20, a global MAF less than 0.01 based on gnomAD, an impact

severity of "HIGH", and had been classified as "exonic". We considered these variants to be potentially damaging. In addition, we excluded from consideration any homozygous-alternate variant that was shared between parent and child as it is unlikely that these variants would be disease causing if observed in a healthy parent [51].

When we queried the GEMINI databases for simple-heterozygous, homozygous-alternate, compound-heterozygous, and *de novo* variants, we used all of the above criteria except for MAF and CADD thresholds. Many of the variants identified as indels or SVs did not have CADD scores or MAF values, most likely due to their rarity in the general population.

Finally, we combined all identified, potentially damaging variants into a single dataset using *R* [52] (*version 4.0.3*) and the *tidyverse packages* [53] (*version 1.3.0*). We identified genes that had at least one child but no more than two children per family with any type of variant (homozygous alternate, simple heterozygous, compound heterozygous, or de novo) identified as part of any type of alteration (SNP, indel, SV).

### Pediatric-cancer predictions

One goal of this study was to evaluate whether we could predict which child in each family had been diagnosed with cancer based solely on DNA variation and the genes in which those variants were identified. As a first step, for each disease, we specified the genes associated with each filtered variant as input to *VarElect* [54]. *VarElect* makes predictions about which gene or genes are likely to affect phenotypes, either directly or indirectly through gene-gene interactions. *VarElect* provides a score for each gene indicating the strength of connection between the gene and phenotypes. In addition, a disease likelihood score is provided for each gene that is based on Gene Damage Index [55] and residual variation intolerance [56] scores. We ranked each gene separately based on these scores. For example, the highest connection score received a rank of one, the second-highest connection score received a rank of two, etc. We ranked the disease likelihood scores using the same logic. We then averaged these two ranked scores for each gene.

To make a prediction about which child had been diagnosed with cancer in each family, we focused on the five genes with the lowest average rank per family. We generated an aggregate score for each child who had at least one variant in the five genes, based on the gene ranks. If a given child had a variant in the top-ranked gene, we increased that child's score by 5; if a child had a variant in the second-ranked gene, we increased that child's score by 4; and so on. If more than one child had a variant in one of the ranked genes, the value for that ranked gene was evenly divided among the samples with a variant in that gene. For example, if two children had a variant in the first ranked gene, each child was given a score of 2.5 for that gene; this logic gave a lower priority to variants that were shared among siblings. We then summed these scores for each child. For example, if a child had a unique variant (not present in any other siblings) in the 1st and 4th-ranked genes, we assigned an aggregate score of 7 to that child. The child with the highest total score in each family was predicted to have been diagnosed with a pediatric tumor.

### Supporting information

**S1 Table. The number of children with a potentially damaging variant per gene per family.** Bolded genes (genes with 2 or less children harboring a potentially damaging variant) were used as part of VarElect analysis.
(XLSX)

**S2 Table. Genes with potentially damaging variants for low-grade astrocytoma.** We included genes that had at least one child but no more than two with a potentially damaging

variant. Scores provided by *VarElect*, our ranked version of the *VarElect* scores, and our overall ranking are shown for each gene. RP11-294C11.1, SPATA31A6, XIRP2, and Y_RNA were also identified in 2 or less children with potentially damaging variants, but were not able to be ranked by *VarElect*. The number of potentially damaging variants in each gene is one unless specified with n. Abbreviations: het = simple heterozygous. SNP = single nucleotide polymorphism. del = deletion. SV = structural variant.
(XLSX)

**S3 Table. Genes with potentially damaging variants for Wilms tumor.** We included genes that had at least one child but no more than two with a potentially damaging variant. Scores provided by VarElect, our ranked version of the VarElect scores, and our overall ranking are shown for each gene. Y_RNA was also identified in 2 or less children with potentially damaging variants, but were not able to be ranked by VarElect. Abbreviations: het = simple heterozygous. hom = homozygous. SNP = single nucleotide polymorphism. del = deletion. SV = structural variant. CH = compound heterozygous.
(XLSX)

**S4 Table. Genes with potentially damaging variants for Burkitt's lymphoma.** We included genes that had at least one child but nor more than two with a potentially damaging variant. Scores provided by VarElect, our ranked version of the VarElect scores, and our overall ranking are shown for each gene. CCDC179, RNU7-167P, and Y_RNA were also identified in 2 or less children with potentially damaging variants, but were not able to be ranked by VarElect. Abbreviations: het = simple heterozygous. hom = homozygous. SNP = single nucleotide polymorphism. del = deletion. SV = structural variant.
(XLSX)

## Acknowledgments

We thank John and Brady Wright from the Mac's Gift Foundation for helping with recruiting families for this study. In addition, we thank the research participants who consented to be part of the study and to share their data.

## Author Contributions

**Conceptualization:** Dustin B. Miller, Stephen R. Piccolo.

**Formal analysis:** Dustin B. Miller.

**Methodology:** Dustin B. Miller.

**Resources:** Reid Robison, Stephen R. Piccolo.

**Supervision:** Stephen R. Piccolo.

**Writing – original draft:** Dustin B. Miller.

**Writing – review & editing:** Dustin B. Miller, Reid Robison, Stephen R. Piccolo.

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
