## [Decision Letter · Decision Letter 0]

23 Aug 2021

PONE-D-21-21297

Toward a methodology for evaluating DNA variants in nuclear families

PLOS ONE

Dear Dr. Piccolo,

Thank you for submitting your manuscript to PLOS ONE. After careful consideration, we feel that it has merit but does not fully meet PLOS ONE’s publication criteria as it currently stands. Therefore, we invite you to submit a revised version of the manuscript that addresses the points raised during the review process.

We look forward to receiving your revised manuscript.

Kind regards,

Alvaro Galli

Academic Editor

PLOS ONE

“Brigham Young University provided startup funds to SRP and a graduate assistantship to DBM.”

 “The author(s) received no specific funding for this work”

Reviewers' comments:

Reviewer's Responses to Questions

**Comments to the Author**

1. Is the manuscript technically sound, and do the data support the conclusions?

Reviewer #1: Yes

2. Has the statistical analysis been performed appropriately and rigorously? 

Reviewer #1: Yes

3. Have the authors made all data underlying the findings in their manuscript fully available?

Reviewer #1: Yes

4. Is the manuscript presented in an intelligible fashion and written in standard English?

Reviewer #1: Yes

5. Review Comments to the Author

Reviewer #1: The manuscript entitled "Toward a methodology for evaluating DNA variants in nuclear families" is a rather interesting approach to evaluate pedriatic cancer risk of children carrying rare variants. Data are exhaustive and the manuscript deserves publication after minor revision just to increase clarity for a non expert reader. Manuscript is well written and methods well explained; however, I recommend some change in the figure legends. Figure 1 needs a more detailed legends. Question marks mean that children have cancer or not? or not sure? Figure two is fine.

From figure 3, I understand that sample ID 29 (from family 1) carries 16 SNPs, 2 indels and 2 SV; totally, 20 potentially damaging variants, right? In table 1, family 1, 1 child carries 12 damaging variants: are these the same data as reported in figure 3? Please explain.

Supplementry tables are exhastive, discussion is fine; however, the authors should point out more convincingly, the importance of the study.

6. PLOS authors have the option to publish the peer review history of their article (what does this mean?). If published, this will include your full peer review and any attached files.

Reviewer #1: No

---

## [Author Response · Author response to Decision Letter 0]

22 Sep 2021

We have prefixed our responses with ">>".

>> We have updated our manuscript to address these requirements. Please let us know if anything else needs to be changed with the formatting.

“Brigham Young University provided startup funds to SRP and a graduate assistantship to DBM.”

“The author(s) received no specific funding for this work”

>> Thank you for letting us know. We have removed our previous statement from the manuscript. Please change our Funding Statement to: "The College of Life Sciences at Brigham Young University provided faculty startup funds to SRP. The Department of Biology at Brigham Young University provided a graduate assistantship to DBM."

>> We are unaware of any references that have been retracted. We went through and checked these in PubMed and did not find any that had been retracted. Please let us know if you are aware of any that we missed.

Reviewer #1:

The manuscript entitled "Toward a methodology for evaluating DNA variants in nuclear families" is a rather interesting approach to evaluate pediatric cancer risk of children carrying rare variants. Data are exhaustive and the manuscript deserves publication after minor revision just to increase clarity for a non expert reader. Manuscript is well written and methods well explained; however, I recommend some change in the figure legends.

>> We thank the reviewer for taking time to review the manuscript and for these encouraging comments.

Figure 1 needs more detailed legends. Question marks mean that children have cancer or not? or not sure? Figure two is fine.

>> We have added clarifying details to the legend for Figure 1, as shown below:

>> "Each diagram in this figure represents one family from our study. In each diagram, shapes at the top represent parents; shapes at the bottom represent children. Numbers below children are sample identifiers. These identifiers do not necessarily correspond to birth order or any other factor. One child in each family had been diagnosed with pediatric cancer, but the disease status of all children was unknown to the primary author until after disease-history predictions were made. This unknown status is represented using question marks."

From figure 3, I understand that sample ID 29 (from family 1) carries 16 SNPs, 2 indels and 2 SV; totally, 20 potentially damaging variants, right? 

>> That is correct. We have added more details to the figure legend to help clarify, as shown below: 

>> "This figure shows the number of potentially damaging variants for each variant type (simple heterozygous, homozygous alternate, and compound heterozygous) and category (SNP, indel, and SV) in each sample. Potentially damaging variants met specific MAF, CADD and impact severity thresholds (see Methods)."

In table 1, family 1, 1 child carries 12 damaging variants: are these the same data as reported in figure 3? Please explain.

>> We apologize for the confusion. Although Table 1 and Figure 3 are complementary, they provide different types of information. Figure 3 illustrates variants that were present in each individual in each family, whereas Table 1 summarizes the overlap in variants among siblings in a given family. Both types of information are important to communicate. We have rewritten the legend for Table 1 to clarify what is shown in that table:

>> "Table 1. An overview of variants in common among children within each nuclear family. We excluded variants that were present in more than two children in a given nuclear family. This table shows the number of variants in common among a specified number of siblings. For example, in Family 1, 12 variants (in 10 genes) were unique to a single child; 5 variants (in 5 genes) were shared between two children. These numbers reflect the number of "potentially damaging" variants—including simple-heterozygous, compound-heterozygous, and homozygous-alternate variants—that met specific MAF, CADD and impact-severity thresholds (see Methods)."

Supplementary tables are exhaustive, discussion is fine; however, the authors should point out more convincingly, the importance of the study.

>> We thank the reviewer for this recommendation. We have added a paragraph at the beginning of the Discussion section that describes additional ways that our study is unique and important:

>> "This study contributes a novel methodology for estimating the translational importance of DNA variants in a given family. Unlike other designs that use trios or quads or that focus primarily on affected family members, we collect data mostly from unaffected individuals, including parents and siblings. Despite the inherent small sample size of comparing individuals within a nuclear family, family members are often exposed to environmental conditions, cultural practices, and dietary habits that are more similar to each other than to the general population, thus potentially reducing confounding factors that could bias an analysis. Although we were not able to demonstrate that our predictions were more accurate than expected by random chance, we have introduced a methodology for identifying candidate variants that can be expanded or refined. Furthermore, we have shared a unique dataset with the community; we are unaware of other WGS datasets from nuclear families with as many as seven children."

>> Later in the Discussion section, we highlight additional factors that make this study unique and important. For example, we explain how our study differs from population-based studies, how we combined quantitative and qualitative methods, the ability to include a wide range of variants (including compound-heterozygous and de novo variants that are not profiled in most genetic studies), our methodology for combining multiple lines of evidence from diverse sources, and insights we gained about potential biases. Finally, even if this manuscript reports some negative results, we are confident that it meets all of the PLOS Publishing Criteria.

>> Thanks again!

---

## [Decision Letter · Decision Letter 1]

27 Sep 2021

Toward a methodology for evaluating DNA variants in nuclear families

PONE-D-21-21297R1

Dear Dr. Piccolo,

We’re pleased to inform you that your manuscript has been judged scientifically suitable for publication and will be formally accepted for publication once it meets all outstanding technical requirements.

Kind regards,

Alvaro Galli

Academic Editor

PLOS ONE

Additional Editor Comments (optional):

Reviewers' comments:

Reviewer's Responses to Questions

**Comments to the Author**

1. If the authors have adequately addressed your comments raised in a previous round of review and you feel that this manuscript is now acceptable for publication, you may indicate that here to bypass the “Comments to the Author” section, enter your conflict of interest statement in the “Confidential to Editor” section, and submit your "Accept" recommendation.

Reviewer #1: All comments have been addressed

2. Is the manuscript technically sound, and do the data support the conclusions?

Reviewer #1: Yes

3. Has the statistical analysis been performed appropriately and rigorously? 

Reviewer #1: Yes

4. Have the authors made all data underlying the findings in their manuscript fully available?

Reviewer #1: Yes

5. Is the manuscript presented in an intelligible fashion and written in standard English?

Reviewer #1: Yes

6. Review Comments to the Author

Reviewer #1: (No Response)

7. PLOS authors have the option to publish the peer review history of their article (what does this mean?). If published, this will include your full peer review and any attached files.

Reviewer #1: No

---

## [Editor Report · Acceptance letter]

30 Sep 2021

PONE-D-21-21297R1 

Toward a methodology for evaluating DNA variants in nuclear families 

Dear Dr. Piccolo:

I'm pleased to inform you that your manuscript has been deemed suitable for publication in PLOS ONE. Congratulations! Your manuscript is now with our production department. 

Kind regards, 

on behalf of

Dr. Alvaro Galli 

Academic Editor

PLOS ONE